# Thermoregulation in Two Models of Trail Run Socks with Different Fabric Separation

**DOI:** 10.3390/life13081768

**Published:** 2023-08-18

**Authors:** Juan Francisco Moran-Cortes, Beatriz Gómez-Martín, Elena Escamilla-Martínez, Raquel Sánchez-Rodríguez, Álvaro Gómez-Carrión, Alfonso Martínez-Nova

**Affiliations:** 1Nursing Department, Universidad de Extremadura (Centro Universitario de Plasencia), Avda. Virgen del Puerto 2, 10600 Plasencia, Spainrsanrod@unex.es (R.S.-R.); podoalf@unex.es (A.M.-N.); 2Nursing Department, Faculty of Nursing, Physiotherapy, and Podiatry, Universidad Complutense de Madrid, 28040 Madrid, Spain

**Keywords:** socks, infrared thermography, trail running, temperature, comfort

## Abstract

Background: Trail running socks with the same fibers and design but with different separations of their three-dimensional waves could have different thermoregulatory effects. Therefore, the objective of this study was to evaluate the temperatures reflected on the sole of the foot after a mountain race with the use of two models of socks with different wave separations. Material and Methods: In a sample of 34 subjects (twenty-seven men and seven women), the plantar temperature was analyzed with the thermal imaging camera Flir E60bx^®^ (Flir systems, Wilsonville, OR, USA) before and after running 14 km in mountainous terrain at a hot temperature of 27 °C. Each group of 17 runners ran with a different model of separation between the waves of the tissue (2 mm versus 1 mm). After conducting the post-exercise thermographic analysis, a Likert-type survey was conducted to evaluate the physiological characteristics of both types of socks. Results: There was a significant increase in temperature in all areas of interest (*p* < 0.001) after a 14 km running distance with the two models of socks. The hallux zone increased in temperature the most after the race, with temperatures of 8.19 ± 3.1 °C and 7.46 ± 2.1 °C for the AWC 2.2 and AWC 3, respectively. However, no significant differences in temperature increases were found in any of the areas analyzed between the two groups. Runners perceived significant differences in thermal sensation between AWC 2.2 socks with 4.41 ± 0.62 points and AWC 3 with 3.76 ± 1.03 points (*p* = 0.034). Conclusion: Both models had a similar thermoregulatory effect on the soles of the feet, so they can be used interchangeably in short-distance mountain races. The perceived sensation of increased thermal comfort does not correspond to the temperature data.

## 1. Introduction

Regular physical exercise has been shown to have beneficial effects on health, making it an integral part of recognized healthy lifestyle habits alongside diet and sleep [1]. The practice of running is related to the improvement and prevention of cardiovascular disease and an increase in longevity in its participants [2,3,4]. During the last 50 years, running has become an easily accessible sporting activity with many practitioners around the world [5,6]. Outdoor activities such as trekking, hiking, and mountain running have gained many followers, experiencing exponential growth in recent years due to their contact with natural environments [7]. The market for shoes, GPS devices, and clothing has adapted to this growth by offering various options for each type of practitioner [8]. One of the most commonly used accessories in sports practice is socks. Thus, both the fibers, the type of knitting or fabric, and the design of the socks are intended to prevent repetitive friction, promote thermoregulation [9], maintain optimal levels of dermal hydration, and facilitate the evacuation of moisture generated by exercise [5]. It has been shown that the use of socks during the race reduces oxygen circulation in the leg, reduces muscle fatigue, and improves muscle oscillation during the run [8]. Chang’s research resulted in an enhancement of ankle proprioception, which could also influence sport biomechanics and have the potential to reduce associated injuries [10]. New configurations of the plantar part of the socks have been shown to be effective in reducing dynamic plantar pressures [11,12,13], providing extra comfort and cushioning for long-distance races. In this regard, current manufacturers offer specific designs based on combinations of fibers (primarily synthetic) that have demonstrated greater comfort and cushioning while keeping the feet well thermoregulated and dry for extended periods [14]. Currently, there is a tendency for the use of three-dimensional socks with plantar waves of non-woven material, creating spaces that facilitate sweat evacuation and proper thermoregulation (Figure 1).

Previous research has shown that wave separations of 5 or 3 mm provide equal thermoregulation, but only in socks designed for short distances (10 km or less) and street races [15]. These socks are thinner and softer due to the fact that as little material as possible is required to minimize weight and provide the best proprioception to suit the quick sporting gesture of the short distance race. Trail or mountain socks must be specifically designed for longer distances, repetitive gestures, irregular terrain, and adaptation to possible climate changes in their progress. Therefore, there is a need to incorporate more material with a narrow wave separation (1 or 2 mm) that allows good adaptation and thermoregulation and with technology to avoid blisters. This configuration is expected by the manufacturer to provide good comfort, cushioning, and, most importantly, a good thermoregulation effect for trail and long-distance races. However, there are no specific studies on mountain-specific socks regarding the effect of this wave separation on the temperature reflected in the sole of the foot after a trail running race. Therefore, the working hypothesis is that wider separations between the 3D waves could provide better thermoregulation in trail running socks. Therefore, the objective of this study is to analyze the temperature changes in the sole of the foot after a 14 km mountain race (a popular running competition distance in our region) using two commercially available socks with the same composition but differing in the separation of their plantar waves.

## 2. Materials and Methods

We designed a short-term, longitudinal, and experimental study. The sample consisted of 34 subjects (27 males and 7 females), with a mean age of 42.06 ± 8.8 years, a mean weight of 69.29 ± 10.58 kg, a mean height of 171 ± 0.07 cm, and a mean body mass index of 23.49 ± 2.71 kg/m^2^ (Table 1). The sample was divided at random into two groups (type 1, Air Waves Comfort, AWC 2.2, n = 17; and type 2, AWC 3, n = 17) according to the model of socks with which they were going to run. No significant differences in the anthropometric variables were found between both groups (Table 2). This study had the approval of the Bioethics Committee of the University of Extremadura, Spain (ID:180//2020), and was planned and carried out following the ethical principles of the Declaration of Helsinki. All participants signed an informed consent statement prior to participating in the study.

The study was conducted with subjects who met the following inclusion criteria: (1) subjects over 18 years old; (2) structurally normal feet, assessed by an experienced podiatrist (last author) as not having non-asymmetrical foot shapes or evident deformities; (3) regular mountain runners (3 or more times a week and/or >40 km weekly). Subjects were excluded if they had: (4) plantar pain; (5) obvious gait or balance abnormalities; (6) inability to run more than 10 km consecutively on the day of the test; or (7) skin abnormalities such as wounds and/or blisters.

Runners were provided with the following guidelines to ensure optimal thermal imaging before the race [16]: (I) thoroughly cleanse the feet; (II) ensure sufficient rest before the examination; (III) avoid direct sunlight exposure; (IV) refrain from applying creams or lotions to the areas being evaluated; (V) avoid consuming hot or cold beverages prior to the examination; (VI) maintain a stable room temperature; (VII) avoid engaging in intense physical activity before the test; (VIII) abstain from smoking and alcohol consumption before the test; (IV) do not use heating devices on the feet; (X) avoid applying chemical products to the feet; and (XI) refrain from using dressings or bandages on the feet.

During the procedure, participants were seated on a medical bed and instructed to remove their socks without touching any surfaces. Their feet were allowed to acclimate to room temperature for one minute. Then, a black thermal screen was placed around the ankle area to prevent heat from other body parts from affecting the feet.

The images were captured in an area free of light reflections with uniform lighting. A properly positioned camera with optimized image settings was mounted on a tripod with a focal length of 1 m from both feet, following the protocol described by Gatt et al. [17].

Next, participants were instructed to place their feet with a separation of 5–10 cm and a slightly upward flexed posture.

Subsequently, a thermographic image was captured using a FLIR E60BX thermal imaging camera with the following technical specifications: a resolution of 76,800 pixels, a thermal sensitivity of 0.045 °C at 30 °C, a temperature range of −20 °C to 120 °C with an accuracy of ±2% or 2 °C, and a spectral range of 7.5 μm to 13 µm.

The images were captured with an emissivity of 0.98 and the ironbow color palette. All measurements were taken on the same day as the trail competition (June 2022), during the same period, and in the absence of adverse weather conditions such as wind or rain (average temperature of the day: 27 °C).

The participants were provided commercially available socks according to the group (AWC 2.2 or AWC 3, Figure 2), one with a configuration of separated wave patterns on the sole and the other with a pattern of closer waves. The Lurbel© (España, Ontinyent, Val) Pro-Line AWC 2.2 (Type 1) and AWC 3 (Type 2) socks (MLS Textiles 1992, Ontinyent, Valencia) were evaluated. Both socks were composed of 50% Regenactiv (cellulosic-based fiber added to ionic chitosan particles), 25% Cool-Tech, 17% ionized Polyamide, and 8% Lycra. Both socks incorporated the AWC (Air Waves Comfort^®^, a commercial trademark of the socks) technology, which consisted of a denser wave pattern on the base fabric of the socks. The difference between the two models was the separation between the waves (Figure 3). With the socks worn on the runners’ feet (size M, 39–42), the AWC 2.2 model had a plantar wave configuration with a separation of 2 mm, while the AWC 3 model had a separation of 1 mm.

The participants ran a 14 km trail (competition of the athletics federation of Extremadura) with an elevation gain of 345 m while wearing socks and their own model of sports shoes. The temperature during the activity was 27 °C. Immediately after the race, a new thermal image was taken, following the same protocol described. ThermoHuman software (https://thermohuman.com/es/) was used for temperature extraction from the thermal images captured. This approach has demonstrated excellent reliability and time-saving benefits compared to manual image analysis [18,19]. The sole of the foot was divided into 9 regions of interest (ROIs) (Figure 4): (a) minor toes; (b) hallux; (c) 5th metatarsal head; (d) 2nd, 3rd, and 4th metatarsal heads; (e) 1st metatarsal head; (f) lateral midfoot; (g) medial midfoot; (h) lateral heel; and (i) medial heel (Figure 3). To mitigate bias, the researcher responsible for thermographic analyses was kept unaware of the study.

To assess the perceived comfort and physiological characteristics of the participants, a Likert scale test with a scoring range of 1–5, commonly used in health studies [19], was employed. The test consisted of 7 questions with varying degrees of agreement and disagreement in Likert format. This allows us to quantify the participants’ experiences regarding comfort and physiological characteristics (Table 3).

### Statistical Analysis

In order to maintain data independence [20], all the variables analyzed corresponded to the participant’s left foot, which was chosen at random. The increase in temperature was calculated (Δ, post-pre temperature). After confirming the normality of the data (Kolmogorov–Smirnov test, *p* > 0.05 in all cases), a descriptive analysis and independent samples *t*-test were conducted to determine changes in temperature, comfort, and physiological characteristics between the two socks after the race. Additionally, overall differences and differences by gender were examined. Statistical analyses were performed using SPSS version 22.0 software (campus license, UEX). A significance level of 5% (*p* < 0.05) was established. The temperature increase was analyzed by subtracting the initial temperature before the activity from the temperature measured after the activity (Δ, post-pre).

## 3. Results

### Temperature by Area Pre and Post

The areas with the highest average temperature were found before the test in the medial arch with 30.22 ± 1.17 and after the race in the same area with 34.58 ± 1.47. The area with the lowest average temperature was hallux, with 24.51 ± 1.78 and 32.34 ± 2.38, respectively, pre and post (Table 4). No significant difference was found between the areas.

No significant differences were found in the temperature increase (Δ, post-pre) between the AWC 2.2 and AWC 3 socks in the studied thermal area (Table 5).

By gender, the highest temperature increase for men was found in the 2–5 toes area with 7.80 ± 2.68, while for women, it was in the hallux area with 8.30 ± 1.70. On the other hand, the area with the lowest temperature increase was the medial arch for both men and women, with 4.42 ± 1.57 and 4.17 ± 1.9, respectively. No significant differences were found between the two groups (Table 6).

In terms of comfort (questions 1 to 4 of the test), subjects rated the fitting characteristics highest with a score of 4.82 ± 0.39 for AWC 2.2 and 4.82 ± 0.39 for AWC 3. The lowest scores were for the sock heights, with 4.65 ± 0.61 for AWC 3 and 4.53 ± 0.62 for AWC 2.2 (Table 7).

For the physiological characteristics (questions 4 to 7 of the test), the highest scores were 4.65 ± 0.61 and 4.56 ± 0.63 for cushioning in both socks. No differences were found in the scores for breathability or overall cushioning between the two socks. A significant difference was found for thermal sensation between sock AWC 2.2 with a score of 4.41 ± 0.62 and sock AWC 3 with a score of 3.76 ± 1.03 (*p* = 0.034, Table 8).

## 4. Discussion

This study aims to analyze the temperature changes in the plantar region of the feet after a 14 km mountain run using two types of socks. The socks used share a design based on waves but differ in the distance between them. Both models resulted in an increase of approximately 5 °C in the average foot temperature after the run. This increase is caused by elevated internal temperatures and accelerated blood flow. Additionally, other factors such as friction with the footwear, shoe occlusion, or ambient temperature contributed to this increase in surface temperature on the plantar skin. This average temperature increase of 5 °C (range, 4–8 °C) appears moderate compared to the findings by Jiménez-Pérez [21], who observed temperature increases of up to 10 °C in the plantar region after a 30 min run, indicating that running conditions, such as terrain type, can influence body temperature variation. This is also demonstrated by the study by Nemati, which showed that at a walking speed of 3 km/h, the plantar region experiences a temperature increase of 6 °C, while at 6 km/h and 9 km/h, the temperature increases reach 8 °C and 11.5 °C, respectively [22]. Other authors also found increasing temperatures during running in different conditions, such as with plantar insoles and different footwear [23,24,25]. In our case, the increase of this 5 °C is considered moderate due to the hard conditions of the trail race, with a mean temperature around 27 °C and a positive difference of 450 m in the 14 km.

In the baseline condition, the coldest areas were the hallux and the smaller toes. However, after the 14 km trail run, these areas experienced the greatest increase in temperature. The passage through irregular, rugged, and challenging terrain generates increased gripping effort and friction throughout the distal forefoot area, which may be related to this overheating. Thus, these areas act as a support point for the foot, helping to maintain balance and absorb impact forces, resulting in greater contact and rubbing with elements such as footwear, socks, or the terrain itself, which may not occur in other types of running on regular surfaces or treadmills [22,24]. It is also necessary to consider that the thermoregulatory elements of the socks do not uniformly cover these anatomical structures, which may result in reduced dissipation capacity, leaving them less exposed to the socks’ thermoregulation mechanism.

The highest average temperature recorded both before and after the race was in the medial arch of the foot. This can be explained by the radiation emitted by the blood flow from the medial branch of the posterior tibial artery, as indicated by authors such as Sun et al. [26]. On the other hand, the area where the temperature increased the least was the medial arch, which showed only a slight increase of around 4 °C after exercise (Table 6). This could be related to the fact that during the running phase, the midfoot arch dynamically contracts and relaxes to absorb impact and propel the body forward. This dynamic movement helps to distribute the forces generated during running more evenly, thereby reducing stress on the foot structures. Additionally, this area is affected by heat convection more than the toe region due to the air space between the insole and the plantar surface. Both perspiration and convection play an important role in cooling the arch area, indicating that conduction is not the only dominant mode of heat transfer. In both cases, the two types of socks may aid in this process as the arrangement of the bands provides efficient dissipation by facilitating the flow of excess moisture and heat between them [22,25,27].

Both types of socks show a similar temperature increase, revealing their effective temperature regulation capacity. Socks with smaller separations between the waves (AWC 3) do not exhibit worse thermoregulation or greater heating than those with larger separations (AWC 2.2). The explanation can be found in the structure and properties of the materials used in the manufacturing of the socks than their design. Additionally, socks also act as a thermal insulation barrier between the feet and the environment. In the case of the analyzed socks, having a 1 or 2 mm difference in band separation allows for better air circulation around the feet. This facilitates more efficient removal of moisture and heat as air can move more freely through space. Adequate air circulation prevents heat buildup and promotes sweat evaporation. In light of the results, it appears that the temperature increase is more dependent on physical exercise and the general thermoregulation process rather than the specific sock used. The type of sock, even with a minimal 1 mm separation, does not have a significant impact on the temperature increase during the run. The socks act more as a protective layer between the foot and its surroundings, but their influence on foot thermoregulation is secondary compared to the physiological mechanisms of the body itself. The incorporation of cooling zones in trail socks, specifically in the toe area, could be beneficial in maintaining an appropriate temperature in that area. These cooling zones could be designed with breathable and highly moisture-absorbent materials, allowing for better ventilation and promoting sweat evaporation. Effective thermoregulation in the toe area can help prevent heat and moisture buildup, thereby reducing the risk of blister formation [28]. By keeping the feet cooler and drier, friction is minimized and excessive moisture is avoided, contributing to the prevention of skin injuries. The textile industry has a major challenge here since the specific design of the sock planting area, which has shown beneficial effects in reducing zonal temperature [29] or dynamic plantar pressure [11], could improve comfort and performance in racing with such an important physical demand.

Regarding sex, no changes in foot temperatures were observed after the race, unlike other studies that found gender differences in thermoregulation [30]. This fact shows that both models of socks can be used equally by men and women, not benefiting any gender by having a more positive effect. This could be another challenge for textile companies to manufacture specific socks for men and women, not only based on design or color, but to achieve better thermoregulatory effects by zones or improved comfort adapted to specific anatomical differences, such as a longer and thinner heel in women [31]. As the number of participants is small to compare men and women, these results must be considered an approach to the matter. It could be interesting to extend the sample of women to assure more accurate conclusions.

Regarding comfort, both groups rated the two models of socks positively, finding no differences between them. However, runners in the AWC 3 group perceived slightly lower thermal comfort than those in the AWC 2.2 group. This sense of comfort is perceived through thermal integration (heat transfer) and tactile sensations (friction and softness) that occur on the skin [32,33]. Nevertheless, this perception did not have a real impact on post-run temperature, so it could be an impression due to socks thickness or the difference between the plantar waves [34]. Additionally, the differences observed could be attributable just to the multiple paired comparisons, having no real impact. With these results and both models being of the same cost, runners can choose either, with the only difference between the two being the thermal sensation. Maybe in cold climates, the AWC 3 sock could be more suitable, and the AWC 2.2 sock could be more suitable for warmer climates.

### Limitations of the Study

As shoes have a wide potential to influence foot temperature and comfort, the main limitation of the study was the inability to control the uniformity of the shoe model used for the race. Although all participants wore specific trail shoes to accompany the specific sports gesture, the different upper materials and fabrics may have slightly influenced the thermoregulation process. The findings of this study should be considered an approximation of the topic, given that the number of participants in the study is too small to compare two groups without a control group.

## 5. Conclusions

It appears that the temperature increase is more dependent on physical exercise and overall thermoregulation processes than the sock used. Both sock types, regardless of the separation between their fabric waves, effectively regulate temperature, keeping the foot relatively cool inside the sports shoe. Although runners perceived lower thermal comfort with the AWC 3 sock, this was not related to the actual post-race temperature.

## Figures and Tables

**Figure 1 life-13-01768-f001:**
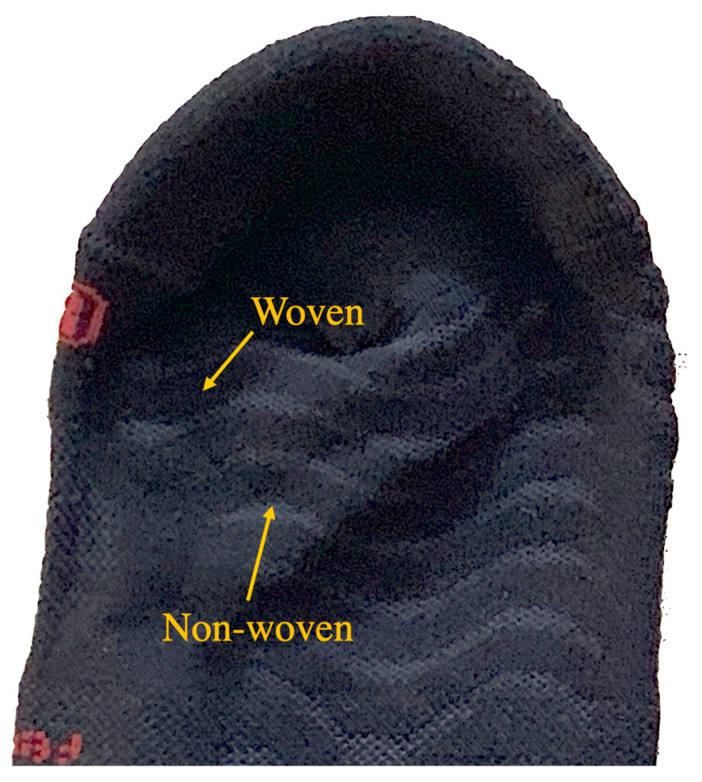
Woven and non-woven 3D waves (2 mm separation).

**Figure 2 life-13-01768-f002:**
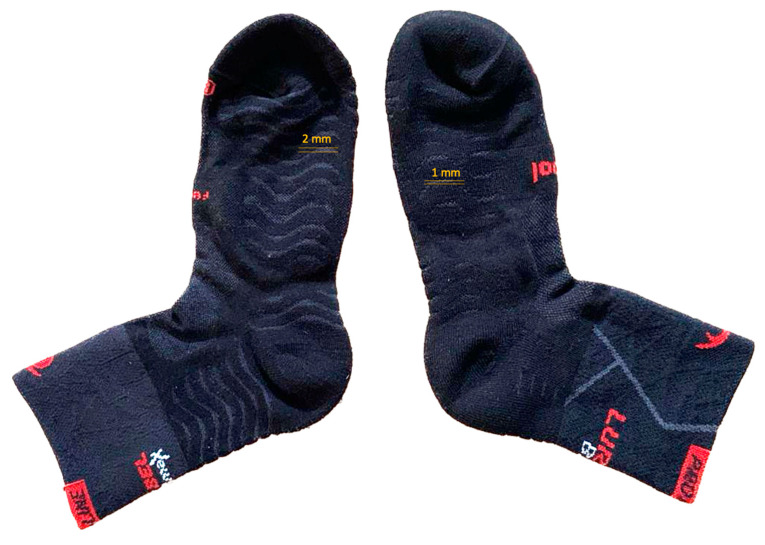
The socks are type 1 or AWC 2.2 (**left**) and type 2 or AWC 3 (**right**).

**Figure 3 life-13-01768-f003:**
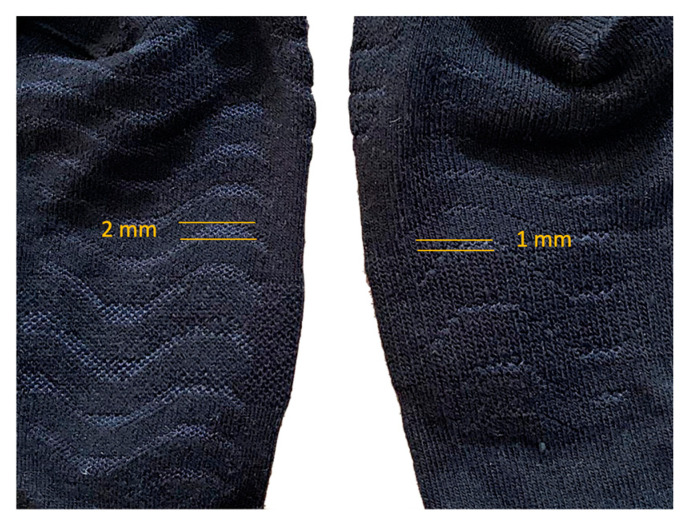
Type of wave separation: 2 mm for the socks on type 1 or AWC 2.2 (**left**) and 3 mm for type 2 or AWC 3 (**right**).

**Figure 4 life-13-01768-f004:**
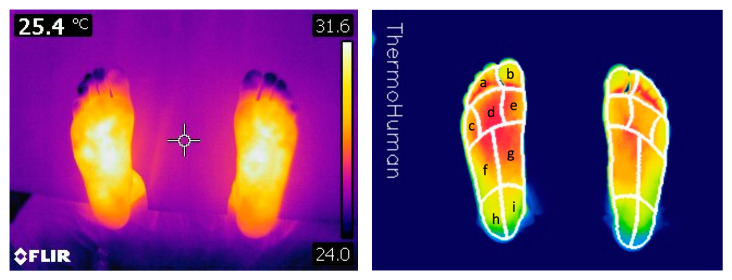
A thermal photograph of a participant’s feet with the analysis of the ROIs.

**Table 1 life-13-01768-t001:** Anthropometric characteristics of the participants.

	n	Minimum	Maximum	Mean	SD
Age	34	22	66	42.06	8.89
Foot size	34	36	45	41.56	1.95
Weight	34	50	89	69.29	10.58
Height	34	1.6	1.87	1.71	0.07
BMI	34	19.2	30.1	23.49	2.71

**Table 2 life-13-01768-t002:** Differences between groups in anthropometric variables. AWC (Air Waves Comfort).

	Type of Sock	N	Mean	SD	*p*
Age	AWC 2.2	17	41.82	9.43	0.880
	AWC 3	17	42.29	8.59	
Foot size	AWC 2.2	17	41.27	2.15	0.386
	AWC 3	17	41.85	1.73	
Weight	AWC 2.2	17	68.65	11.2	0.727
	AWC 3	17	69.94	10.23	
Height	AWC 2.2	17	1.72	0.07	0.635
	AWC 3	17	1.71	0.07	
BMI	AWC 2.2	17	23.13	2.99	0.439
	AWC 3	17	23.86	2.44	

**Table 3 life-13-01768-t003:** Physiological characteristics of the participants.

Comfort										
	(1 Very uncomfortable–5 Very comfortable)									
	AWC1						AWC2			
Sock height	1	2	3	4	5	1	2	3	4	5
Fit/Adaptation	1	2	3	4	5	1	2	3	4	5
Softness/Feel	1	2	3	4	5	1	2	3	4	5
Comfort (overall)	1	2	3	4	5	1	2	3	4	5
Physiological characteristics										
	(1 Very wet–5 Very dry)									
	AWC1						AWC2			
Moisture/Breathability	1	2	3	4	5	1	2	3	4	5
	(1 Very hot–5 Very cool)									
Thermal sensation.	1	2	3	4	5	1	2	3	4	5
	(1 Little cushioned–5 Very cushioned)									
Overall cushioning.	1	2	3	4	5	1	2	3	4	5

**Table 4 life-13-01768-t004:** Average temperature by area pre- and post-intervention.

Area	Pre		Post	
	Mean	SD	Mean	SD
2nd–5th Toes	25.07	1.88	32.74	2.08
Hallux	24.51	1.78	32.34	2.38
5th CMT	27.34	1.58	33.13	1.82
2nd–5th CMT	27.74	1.67	33.45	2.04
1st CMT	27.23	1.57	33.25	2.12
Lateral Arch	29.08	1.42	33.87	1.63
Medial Arch	30.22	1.17	34.58	1.47
Ext Heel	27.81	1.39	33.29	1.97
Int Heel	28.42	1.27	33.68	1.96

**Table 5 life-13-01768-t005:** Average temperature increase by area and socks type.

Área	Socks	N	Mean	SD	*p*
2nd–5th Toes	AWC 2.2	17	7.79	2.97	0.789
	AWC 3	17	7.55	2.09	
Hallux	AWC 2.2	17	8.19	3.19	0.435
	AWC 3	17	7.46	2.08	
5th CMT	AWC 2.2	17	5.81	2.31	0.984
	AWC 3	17	5.79	1.70	
2nd–5th CMT	AWC 2.2	17	5.88	2.48	0.629
	AWC 3	17	5.53	1.66	
1st CMT	AWC 2.2	17	6.38	2.63	0.359
	AWC 3	17	5.66	1.84	
Lateral Arch	AWC 2.2	17	4.75	2.00	0.912
	AWC 3	17	4.82	1.72	
Medial Arch	AWC 2.2	17	4.62	1.78	0.373
	AWC 3	17	4.12	1.46	
Ext Heel	AWC 2.2	17	5.67	2.59	0.625
	AWC 3	17	5.30	1.68	
Int Heel	AWC 2.2	17	5.56	2.46	0.415
	AWC 3	17	4.96	1.65	

**Table 6 life-13-01768-t006:** Average temperature increases by area pre- and post-intervention.

Área	Sex	N	Mean	SD	*p*
2nd–5th Toes	Male	27	7.80	2.68	0.558
	Female	7	7.16	1.90	
Hallux	Male	27	7.71	2.89	0.606
	Female	7	8.30	1.70	
5th CMT	Male	27	5.74	2.08	0.743
	Female	7	6.02	1.77	
2nd–5th CMT	Male	27	5.66	2.15	0.797
	Female	7	5.89	1.96	
1st CMT	Male	27	5.88	2.34	0.485
	Female	7	6.56	2.01	
Lateral Arch	Male	27	4.77	1.85	0.950
	Female	7	4.82	1.93	
Medial Arch	Male	27	4.42	1.57	0.725
	Female	7	4.17	1.94	
Ext Heel	Male	27	5.31	2.25	0.343
	Female	7	6.19	1.69	
Int Heel	Male	27	5.14	2.12	0.521
	Female	7	5.72	2.02	

**Table 7 life-13-01768-t007:** Average comfort characteristics by sock type.

COMFORT	Sock	Mean	SD	*p*
Height	AWC 2.2	4.53	0.62	0.581
	AWC 3	4.65	0.61	
Fit/Adaptation	AWC 2.2	4.82	0.39	0.683
	AWC 3	4.76	0.44	
Touch/Feel	AWC 2.2	4.76	0.44	0.683
	AWC 3	4.82	0.39	
Comfort	AWC 2.2	4.82	0.39	1
	AWC 3	4.82	0.39	
Comfort (overall)	AWC 2.2	4.74	0.26	0.785
	AWC 3	4.76	0.36	

**Table 8 life-13-01768-t008:** Physiological characteristics by type of socks.

Physiological Characteristics	Sock	Mean	SD	*p*
Moisture/Breathability	AWC 2.2	4.29	0.85	0.551
	AWC 3	4.12	0.86	
Thermal Sensation	AWC 2.2	4.41	0.62	0.034
	AWC 3	3.76	1.03	
Overall Cushioning	AWC 2.2	4.56	0.63	0.697
	AWC 3	4.65	0.61	
	AWC 3	4.175	0.66	

## Data Availability

Data availability can be found at www.unex.es.

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
