# Peer review of "Thermoregulation in Two Models of Trail Run Socks with Different Fabric Separation"

_life, 2023, doi:10.3390/life13081768_

Round 1
Reviewer 1 Report
The manuscript generally is written in a way, that made the objective, overall design of the research and results clear. Nevertheless, there is numerous flaws in the details.
The cited references are mostly recent publications (within the last 5 years), with about 1/4 of the references being form older sources, bout they are relevant. The number of self-citations is appropriate, except of one principal point. Nearly the same author group published the paper of very close content in December 2022. (Sánchez-Rodríguez R. Gómez-Martín B, Escamilla-Martínez E, Morán-Cortés JF, Martínez-Nova A. Thermal Response in Two Models of Socks with Different 3-D Weave Separations. Appl. Sci. 2023, 13(1), 71; https://doi.org/10.3390/app13010071). Nevertheless, this paper was not cited neither in introduction, nor in discussion sections.
The presence of this previously published paper jeopardizes the scientific novelty of the present research, as it is not clear, what is the added value of the new research, comparing with previous one. The Introduction section should address this issue. This could be critical for the scientific merits of the paper.
Alongside, the Introduction lack of clarification on what is the features or requirements for mountain trail-specific socks (i.e why they are specific, how they differ from other socks, e.g., for football or park jogging) and the justification why the research on the effect of 3d wave structures separation on the temperature is important. Why could the separation of 3D structures affect the thermoregulation? And, finally, what is the authors’ hypothesis? More details about the socks of interest could be added, too, e,g, are these compression socks?
The experimental design is generally appropriate test the implicit hypothesis of the paper, nevertheless the material and method section have inaccuracies that make estimation of the adequacy of the method difficult.
The two groups, mentioned in the text in line 67 seems to be groups, testing different models of the socks, but it is not clearly described.
Form the description in lined 103-105 is not clear, weather each participant was provided with two different models, so the different feet wore different types, or there was one group with first type of socks, and another - with second type?
If two socks’ models were tested by different groups, did the participants wore the same model of shoes? As the shoes may play way more important role in thermoregulation and comfort feeling, this issue should be addressed in the description of experiment.
Why was the distance of 14 km selected? The selection of the distance is more related to the research method, not to the objective. It could be better to describe a distance in method section, with some arguments on why this distance was chosen.
It is not clear, what is an importance of the sock height parameter (Table 3 )? How is it related with the objective of the study?
Subsection 2.1 dies not mention what was the test, used for the comparison of Likert scale estimation of the comfort and physical sensation?
In line 92, what does correct focal length mean? Just being in focus?
The figures and tables are generally, appropriate and show data properly, except of the table 4. – there is the difference pre- and – post only, regardless the model of the socks, that is in contradiction with the table caption.
Alongside, the ROI numbering could be added to the Figure 3.
Probably, authors could use colour correction for the figures 1 and 2 to make the waves on the socks more visible.
Concerning the results itself, authors could estimate/discuss whether significant difference in thermal sensation may be just the result of multiple paired comparisons (it is difficult to derive from the paper, as no information is given about how Likert scale estimates were compared).
The conclusions are consistent with the evidence and arguments presented.
The ethic issues of the study are properly addressed.
Turning back to the clearness of the text, some comments could be added:
From the sentence In lines 47 – 49 it is not clear, how the results of Chang (10), who explored effect of the compression socks is related to the reference (11), where effect of cream is studied. The sentence is not well – written – skipping the first clause, one got “Chang’s research (10) resulted … in a relationship with improved biomechanics and sport… “. The relationship either exists or not, regardless of whether the research has been made. Alongside, what does “ improvement in sport” mean?
The sentence 55-57 is not clear – is it about use of socks with three-dimensional structure, or about incorporation of unwoven materials in the socks. Or about incorporation of 3D wavy patterns and unwoven materials in socks?
As the objective is specific and narrow, not always the reader may understand the context. It could be recommended to have reference to the socks image already at the introduction, to illustrate, how these 3D structures look like
The abbreviation AWC appears before (Table 1) it have been described in the text.
In the lines 79 – 86, it is better use one bracket for the list, like 1), just not to mix list item numbers with references.
Consider use “..both models of socks” in line 21
Consider the use of “socks” instead of “sock” in introduction and onward.
Consider “use” instead of “incorporation” in line 55.
There are spaces missed in the text (i.e line 41 line 45)
Author Response
Dear Reviewer´s
Thank you for your kind and very helpful comments. I have looked through the manuscript following your recommendations and suggestions.
I will show to you the changes we have made due to your review, which are underlined in yellow in the paper.
Reviewer #1:
The manuscript generally is written in a way, that made the objective, overall design of the research and results clear. Nevertheless, there is numerous flaws in the details.
Thank you for your comments
The cited references are mostly recent publications (within the last 5 years), with about 1/4 of the references being form older sources, but they are relevant. The number of self-citations is appropriate, except of one principal point. Nearly the same author group published the paper of very close content in December 2022. (Sánchez-Rodríguez R. Gómez-Martín B, Escamilla-Martínez E, Morán-Cortés JF, Martínez-Nova A. Thermal Response in Two Models of Socks with Different 3-D Weave Separations. Appl. Sci. 2023, 13(1), 71; https://doi.org/10.3390/app13010071). Nevertheless, this paper was not cited neither in introduction, nor in discussion sections. The presence of this previously published paper jeopardizes the scientific novelty of the present research, as it is not clear, what is the added value of the new research, comparing with previous one. The Introduction section should address this issue. This could be critical for the scientific merits of the paper.
Response: We have added a new information about this issue. "In this way, a previous research has shown that wave separations of 5 or 3 mm performs equally thermoregulation, but in socks designed for short distance (10 km or less) and street races " and we have added the reference.
Our previous research was made in short distance socks and designed for street races, not trail ones. Also, this previous socks (AWC 1 and AWC 2.1) was different, with wave separations of 5 and 3 mm and in the current research this separations are of 2 (AWC 2.2) and 1 mm (AWC 3). So, we think that there is novelty in the research.
Alongside, the Introduction lack of clarification on what is the features or requirements for mountain trail-specific socks (i.e why they are specific, how they differ from other socks, e.g., for football or park jogging) and the justification why the research on the effect of 3d wave structures separation on the temperature is important. Why could the separation of 3D structures affect the thermoregulation? And, finally, what is the authors’ hypothesis? More details about the socks of interest could be added, too, e,g, are these compression socks?
Response: We have added new info about these issues, with the specific characteristics of the trail socks and the needing of it. We have added to the working hypothesis.
The experimental design is generally appropriate test the implicit hypothesis of the paper, nevertheless the material and method section have inaccuracies that make estimation of the adequacy of the method difficult.
The two groups, mentioned in the text in line 67 seems to be groups, testing different models of the socks, but it is not clearly described.
Response: we have added the groups information in the materials and methods seccition
Form the description in lined 103-105 is not clear, weather each participant was provided with two different models, so the different feet wore different types, or there was one group with first type of socks, and another - with second type?
Response: we have clarified this issue adding new info.
If two socks’ models were tested by different groups, did the participants wore the same model of shoes? As the shoes may play way more important role in thermoregulation and comfort feeling, this issue should be addressed in the description of experiment.
Response: we have added that runners run with his own model of shoes. As this could be a limitation, this has been added to the limitation section.
Why was the distance of 14 km selected? The selection of the distance is more related to the research method, not to the objective. It could be better to describe a distance in method section, with some arguments on why this distance was chosen.
Response: this distance was selected because was a trail competition of the athletics of Extremadura.
It is not clear, what is an importance of the sock height parameter (Table 3 )? How is it related with the objective of the study?
Response: we think that in important to assess the feelings of the runners in this issue, although is not the main goal of our study
Subsection 2.1 dies not mention what was the test, used for the comparison of Likert scale estimation of the comfort and physical sensation?
Response: We have added new info about these issues
In line 92, what does correct focal length mean? Just being in focus?
Response: we have clarified this issue
The figures and tables are generally, appropriate and show data properly, except of the table 4. – there is the difference pre- and – post only, regardless the model of the socks, that is in contradiction with the table caption.
Response: we have modified this issue
Alongside, the ROI numbering could be added to the Figure 3.
Response: we have modified this figure
Probably, authors could use colour correction for the figures 1 and 2 to make the waves on the socks more visible.
Response: we have modified this figure with colour and wider size
Concerning the results itself, authors could estimate/discuss whether significant difference in thermal sensation may be just the result of multiple paired comparisons (it is difficult to derive from the paper, as no information is given about how Likert scale estimates were compared).
Response: we have added this issue
The conclusions are consistent with the evidence and arguments presented.
Response: thank you for your gentle comment.
The ethic issues of the study are properly addressed.
Response: thank you for your gentle comment.
Turning back to the clearness of the text, some comments could be added:
From the sentence In lines 47 – 49 it is not clear, how the results of Chang (10), who explored effect of the compression socks is related to the reference (11),
where effect of cream is studied. The sentence is not well – written – skipping the first clause, one got “Chang’s research (10) resulted … in a relationship with improved biomechanics and sport… “. The relationship either exists or not, regardless of whether the research has been made. Alongside, what does “ improvement in sport” mean?
Response: we have clarified this issue
The sentence 55-57 is not clear – is it about use of socks with three-dimensional structure, or about incorporation of unwoven materials in the socks. Or about incorporation of 3D wavy patterns and unwoven materials in socks?
Response: we have clarified this issue as this tendence is to at not woven materials
As the objective is specific and narrow, not always the reader may understand the context. It could be recommended to have reference to the socks image already at the introduction, to illustrate, how these 3D structures look like
Response: we have added the new figure in the introduction.
The abbreviation AWC appears before (Table 1) it have been described in the text.
Response: we have clarified this issue
In the lines 79 – 86, it is better use one bracket for the list, like 1), just not to mix list item numbers with references.
Response: we have changed this list
Comments on the Quality of English Language
Consider use “..both models of socks” in line 21
Response: we have changed this issue
Consider the use of “socks” instead of “sock” in introduction and onward.
Response: we have changed this issue
Consider “use” instead of “incorporation” in line 55.
Response: we have changed this issue
There are spaces missed in the text (i.e line 41 line 45)
Response: we have changed this issue
Reviewer 2 Report
Dear authors:
In order to improve your manuscript, I would make these suggestions:
“Materials and Methods” section
1. How do you verify runners have a “structurally normal feet” as an inclusión criteria? (line 75).
2. The text is clear and easy to read, however the “Materials and Methods” section is a bit complicated. I would suggest using different paragraphs for the guidelines provided to participants and using Roman numerals or letters, as It can be confusing with the number of bibliographic references (line 80-86). So, are these all requirements from reference (16)? What does the number 16 mean?
This suggestion is valid from text between line 118-121.
3. Also, the research procedure should be explained more clearly (line 86-122). I would start by separating sentences and using punctuation marks (period, full stop and commas).
4. Actually, I miss a control group, since I understand that it is no longer possible to include it, I would at least consider it a big limitation, specially when you state (line 219-221): At the light of the results, it appears that the temperature increase is dependent on physical exercise and the general thermoregulation process rather than the specific sock used.
5. Could you add a sentence about the design of the study?
“Results” section
1. I find very interesting this part. However no reference to laterality is mentioned. The tables 4,5 and 6 are not well explained considering that you are including both feet. is not it n=68?. Is it the average of the data between the rigth and the left limb of each participant (so n=34). I think you should include this fact.
“Discussion” section
1. Good part. I would suggest including a paragraph on cost-benefit to choose any model of sock. I mean, I think the bottom line is that both models are useful, but are they both the same cost to the customer? Since there is no conflict of interest, that could improve the discussion of the manuscript (only socks were delivered for the Company to carry out the study)
“Limitation” section
1. I would include these concerns in some way as weaknesses of the study:
The number of participants is small to compare two groups without control one. The gender comparison is necessary, however the outcome of this study must be considered an approach to the matter. Specially when you write (Line 236-239): Regarding sex, no changes in foot temperatures were observed after the race, unlike other studies that found gender differences in thermoregulation (31). This fact shows that both models of socks can be used equally by men and women, not benefiting any gender…)
Author Response
Thank you for your kind and very helpful comments. I have looked through the manuscript following your recommendations and suggestions.
I will show to you the changes we have made due to your review, which are underlined in green in the paper.
Reviewer 2
Dear authors:
In order to improve your manuscript, I would make these suggestions:
“Materials and Methods” section
- How do you verify runners have a “structurally normal feet” as an inclusion criteria? (line 75).
Response: we have added new info about these issue. The feet was assessed by a experimented podiatrist (last author)
- The text is clear and easy to read, however the “Materials and Methods” section is a bit complicated. I would suggest using different paragraphs for the guidelines provided to participants and using Roman numerals or letters, as It can be confusing with the number of bibliographic references (line 80-86). So, are these all requirements from reference (16)? What does the number 16 mean? This suggestion is valid from text between line 118-121.
Response: we have changed this issues, and the requirements are extracted from the reference 16. This has been clarified in the text
- Also, the research procedure should be explained more clearly (line 86-122). I would start by separating sentences and using punctuation marks (period, full stop and commas).
Response: we have changed this issues,
- Actually, I miss a control group, since I understand that it is no longer possible to include it, I would at least consider it a big limitation, specially when you state (line 219-221): At the light of the results, it appears that the temperature increase is dependent on physical exercise and the general thermoregulation process rather than the specific sock used.
Response: we have added this issue in limitation section
- Could you add a sentence about the design of the study?
Response: we have added this issue in materials and method section
“Results” section
- I find very interesting this part. However no reference to laterality is mentioned. The tables 4,5 and 6 are not well explained considering that you are including both feet. is not it n=68?. Is it the average of the data between the rigth and the left limb of each participant (so n=34). I think you should include this fact.
Response: we have considered as our sample is 34, as we consider that to maintain the independence of the data, only one foot must be entered in the statistic analysis. This sentence has been added to the text and also the reference.
“Discussion” section
- Good part. I would suggest including a paragraph on cost-benefit to choose any model of sock. I mean, I think the bottom line is that both models are useful, but are they both the same cost to the customer? Since there is no conflict of interest, that could improve the discussion of the manuscript (only socks were delivered for the Company to carry out the study)
Response: we have added this issue about the cost-benefit in discussion section
“Limitation” section
- I would include these concerns in some way as weaknesses of the study:
The number of participants is small to compare two groups without control one. The gender comparison is necessary, however the outcome of this study must be considered an approach to the matter. Specially when you write (Line 236-239): Regarding sex, no changes in foot temperatures were observed after the race, unlike other studies that found gender differences in thermoregulation (31). This fact shows that both models of socks can be used equally by men and women, not benefiting any gender…)
Response: we have added a text with an explanation of these issue. It could be interesting to expand the sample of women to assure better results
Reviewer 3 Report
Introduction: In the present paper, the authors investigated foot temperature management and comfort in commercially available socks with AWC (Advanced Wave Control) technology, donated by the company MLS Textiles 265 in 1992. The study aimed to compare the effects of different densities of knitted waves in the sole part of the sock during mountain race.
Before delving into a detailed review, I would like to raise several conceptual questions concerning the presented paper:
-
1.The authors previously published a paper titled "Thermal Response in Two Models of Socks with Different 3-D Weave Separations" (Applied Sciences, 13(1), 71) which shares similarities with the current research in terms of scope and results. Surprisingly, the authors did not cite this prior work in their present publication. It is essential for the authors to clarify the differences between the two papers in terms of aims and scientific novelty.
-
2. The paper lacks an explanation for the specific choice of socks with wave distances of 2 and 3 mm, as opposed to socks with a broader range of distances. In their previous research, the distances were 3 and 5 mm. To enhance the study's credibility, it would be beneficial to provide a rationale for this selection.
-
3. The research methodology does not address how the authors accounted for the potential influence of shoes on the measured foot temperature and comfort. It is crucial to acknowledge that shoes can significantly impact both comfort and thermoregulation properties. Thus, addressing this potential confounding factor is necessary for a comprehensive analysis.
Despite these issues, if the above-mentioned concerns are adequately addressed, the paper could be of considerable interest to sock consumers. The study effectively demonstrates that AWC technology does not offer significant benefits compared to regular sport socks, and highlights potential marketing tactics used by manufacturers. The authors correctly emphasize the importance of exploring more effective materials and scientifically validated sock designs to enhance the beneficial properties of socks for various sports.
Author Response
Dear Reviewer
Thank you for your kind and very helpful comments. I have looked through the manuscript following your recommendations and suggestions.
I will show to you the changes we have made due to your review, which are underlined in blue in the paper.
Reviewer 3
Introduction: In the present paper, the authors investigated foot temperature management and comfort in commercially available socks with AWC (Advanced Wave Control) technology, donated by the company MLS Textiles 265 in 1992. The study aimed to compare the effects of different densities of knitted waves in the sole part of the sock during mountain race.
Before delving into a detailed review, I would like to raise several conceptual questions concerning the presented paper:
- The authors previously published a paper titled "Thermal Response in Two Models of Socks with Different 3-D Weave Separations" (Applied Sciences, 13(1), 71) which shares similarities with the current research in terms of scope and results. Surprisingly, the authors did not cite this prior work in their present publication. It is essential for the authors to clarify the differences between the two papers in terms of aims and scientific novelty.
Response: We have added a new information about this issue.
In this way, a previous research has shown that wave separations of 5 or 3 mm performs equally thermoregulation, but in socks designed for short distance (10 km or less) and street races.
Also, we have added the reference of our previous work.
We think that the current paper adds insights to choose the better sock for trail and mountain races … as the separation of the waves in this work are 2 mm and 1 mm, more closet than in the previous paper.
- The paper lacks an explanation for the specific choice of socks with wave distances of 2 and 3 mm, as opposed to socks with a broader range of distances. In their previous research, the distances were 3 and 5 mm. To enhance the study's credibility, it would be beneficial to provide a rationale for this selection.
Response: We have a more expanded paragraph in the text (before figure 2, line 138 aprox)
- The research methodology does not address how the authors accounted for the potential influence of shoes on the measured foot temperature and comfort. It is crucial to acknowledge that shoes can significantly impact both comfort and thermoregulation properties. Thus, addressing this potential confounding factor is necessary for a comprehensive analysis.
Response: This issue was not assessed in pour work, so due to the importance of it, we have added a text with this issue in the limitations section
Despite these issues, if the above-mentioned concerns are adequately addressed, the paper could be of considerable interest to sock consumers. The study effectively demonstrates that AWC technology does not offer significant benefits compared to regular sport socks, and highlights potential marketing tactics used by manufacturers. The authors correctly emphasize the importance of exploring more effective materials and scientifically validated sock designs to enhance the beneficial properties of socks for various sports.
Response: thank you for the your kind comment
Round 2
Reviewer 1 Report
The paper was significantly improved. The questions concerning the materials and methods are generally, properly addressed. Nevertheless, some questions on the motivation and hypothesis remains:
Lines 62-66: the connection to the previous study is clear now. But some comments why socks with wider separation are appropriate for short runs but not for the long runs still is not clear. (Because of this the hypothesis looks controversial: if one expects that the wider separation is better, why this study rejects socks with separation 3-5 mm.)
Line 79: AWC or Air Wave Control still is not explained. Is it a trademark of the socks? It is nothing bad to mention in the introduction that the research was made using commercial socks – see also comment to the line 130-131.
Lines 130 -131. One could recommend adding this the info about the origin of the socks to introduction, specifying that the object of the research is commercially available socks – see also the comment to the line 79.
Lines 138 – 141 – consider moving this part to introduction, see also previous comments
Line 72: authors could add a note, that 14 km is a popular running competition distance in their region (As it was nicely explained in the comments to reviewer) already here, or just not mention the length of the distance (it is, actually, mentioned in the method section) .
Some comments on selection of the experiment design would be good for the paper: why the design when each participant got pair of socks of the certain type was selected, in contrast to the author’s previous research, when each participant received two socks of different types for the left and right leg?
Alongside, there are some language comments:
Lines 62-66 The style of the text could be improved. E.g. “separations of 5 or 3 mm perform equally for thermoregulation…” or “separations of 5 or 3 mm provide equal thermoregulation..” (but not limited to this) .
Line 22: please, change to “for the both types …” or “of the both types…”
Line 48: better “sport biomechanics”;
Line 62: “In this way” is redundant.
Line 91-92: assessed by experimented podiatrist (last author) as non-asymmetrical foot shapes or evident deformities – does it mean “as not having non-asymmetrical foot shapes or evident deformities”?
Line 128: Better: “The participants were provided with the proper socks…
Comments to Figures:
Line 58: add reference to Fig.1 (if you decide to keep it, see next comments figures
Figure 2 is OK now. The waves are clearly visible, and Fig. 1 is redundant. Consider merging of Fig.1 and either Fig.2 or Fig.3.
Consider reducing of the size of the Fig. 3.
Author Response
Dear Reviewer´s
Thank you for your kind and very helpful comments. I have looked through the manuscript following your recommendations and suggestions.
I will show to you the changes we have made due to your review, which are underlined in yellow in the paper.
Reviewer #1:
The paper was significantly improved. The questions concerning the materials and methods are generally, properly addressed. Nevertheless, some questions on the motivation and hypothesis remains:
Thank you for your comments
Lines 62-66: the connection to the previous study is clear now. But some comments why socks with wider separation are appropriate for short runs but not for the long runs still is not clear. (Because of this the hypothesis looks controversial: if one expects that the wider separation is better, why this study rejects socks with separation 3-5 mm.)
Response: We have expanded this introduction section
Line 79: AWC or Air Wave Control still is not explained. Is it a trademark of the socks? It is nothing bad to mention in the introduction that the research was made using commercial socks – see also comment to the line 130-131.
Response: we have clarified this issue adding new info in introduction section
Lines 130 -131. One could recommend adding this the info about the origin of the socks to introduction, specifying that the object of the research is commercially available socks – see also the comment to the line 79.
Response: we have modified this issue adding new info in line 128 “The participants were provided of commercially available socks the proper socks according to the group…”
Lines 138 – 141 – consider moving this part to introduction, see also previous comments
Response: we have moved this part to introduction
Line 72: authors could add a note, that 14 km is a popular running competition distance in their region (As it was nicely explained in the comments to reviewer) already here, or just not mention the length of the distance (it is, actually, mentioned in the method section) .
Response: we have eliminated this data of the introduction as it is redundant in the methodology
Some comments on selection of the experiment design would be good for the paper: why the design when each participant got pair of socks of the certain type was selected, in contrast to the author’s previous research, when each participant received two socks of different types for the left and right leg?
Response: The design of our study differs from the previous study, as in this one, we aimed to compare two groups with a specific type of sock, rather than one foot against another.
Alongside, there are some language comments:
Lines 62-66 The style of the text could be improved. E.g. “separations of 5 or 3 mm perform equally for thermoregulation…” or “separations of 5 or 3 mm provide equal thermoregulation..” (but not limited to this) .
Response: we have added “..separations of 5 or 3 mm provide equal thermoregulation. ”
Line 22: please, change to “for the both types …” or “of the both types…”
Response: we have changed “for the both types..” in the tex
Line 48: better “sport biomechanics”;
Response: we have changed “sport biomechanics..” in the tex
Line 62: “In this way” is redundant.
Response we have considered eliminating redundancy
Line 91-92: assessed by experimented podiatrist (last author) as non-asymmetrical foot shapes or evident deformities – does it mean “as not having non-asymmetrical foot shapes or evident deformities”?
Response: we have replaced for a better understanding “as not having non-asymmetrical foot shapes or evident deformities”
Line 128: Better: “The participants were provided with the proper socks…
Response: we have taken your suggestion into consideration in the text
Comments to Figures:
Line 58: add reference to Fig.1 (if you decide to keep it, see next comments figures
Response: We have taken into consideration your contribution
Figure 2 is OK now. The waves are clearly visible, and Fig. 1 is redundant. Consider merging of Fig.1 and either Fig.2 or Fig.3.
Response: the figure is a contribution made by another reviewer
Consider reducing of the size of the Fig. 3.
Response: we have reduced figure 3 in half.

Reviewer 3 Report
Introduction: The paper has undergone significant improvements, and many drawbacks have been eliminated. However, there are still some areas that require clarification:
-
The authors hypothesized that "wider separations between the 3D waves could provide better thermoregulation" and used 1- and 2-mm separations between waves. In the previous paper, the separation was larger - 3 and 5 mm. It is unclear why the authors chose socks with smaller wave separation for this study. The contradiction between the hypothesis and the research design should be addressed. It is necessary to specify the rationale behind selecting sock models with 1 and 2mm separation. Are these socks intended for special mountain trail use, while 3 and 5mm separations are meant for general running socks?
-
The relevance of reference (11) is unclear in the context of the present paper. It does not seem to contribute to the topic at hand, and thus, its inclusion should be justified or reconsidered.
Overall, the introduction has been improved, but addressing these specific points will enhance the clarity and focus of the paper further.
Author Response
Dear Reviewer´s
Thank you for your kind and very helpful comments. I have looked through the manuscript following your recommendations and suggestions.
I will show to you the changes we have made due to your review, which are underlined in blue in the paper.
Reviewer #3:
Introduction: The paper has undergone significant improvements, and many drawbacks have been eliminated. However, there are still some areas that require clarification:
Thank you for your comments
1.- The authors hypothesized that "wider separations between the 3D waves could provide better thermoregulation" and used 1- and 2-mm separations between waves. In the previous paper, the separation was larger - 3 and 5 mm. It is unclear why the authors chose socks with smaller wave separation for this study. The contradiction between the hypothesis and the research design should be addressed. It is necessary to specify the rationale behind selecting sock models with 1 and 2mm separation. Are these socks intended for special mountain trail use, while 3 and 5mm separations are meant for general running socks?
Response: We have exerted effort to expound that broader wave separations are intended for higher-velocity races, whereas narrower-spaced waves, characterized by increased material density and cushioning, are tailored for mountain racing events.
2.- The relevance of reference (11) is unclear in the context of the present paper. It does not seem to contribute to the topic at hand, and thus, its inclusion should be justified or reconsidered.
Response: the reference has been deleted following your suggestion
Overall, the introduction has been improved, but addressing these specific points will enhance the clarity and focus of the paper further.
Response: Thank you for your words of encouragement.